# Revisiting a Core–Jet Laboratory at High Redshift: Analysis of the Radio Jet in the Quasar PKS 2215+020 at $z = 3.572$

Sándor Frey [1,2,3,*], Judit Fogasy [1,2], Krisztina Perger [1,2], Kateryna Kulish [4], Petra Benke [5,6], Dávid Koller [1,2,7] and Krisztina Éva Gabányi [1,2,7,8]

1   Konkoly Observatory, HUN-REN Research Centre for Astronomy and Earth Sciences, Konkoly Thege Miklós út 15-17, H-1121 Budapest, Hungary; fogasy.judit@csfk.org (J.F.); perger.krisztina@csfk.org (K.P.); koller.david@csfk.org (D.K.); k.gabanyi@astro.elte.hu (K.É.G.)

2   CSFK, MTA Centre of Excellence, Konkoly Thege Miklós út 15-17, H-1121 Budapest, Hungary

3   Institute of Physics and Astronomy, ELTE Eötvös Loránd University, Pázmány Péter sétány 1/A, H-1117 Budapest, Hungary

4   Department of Astronomy, Physics of the Earth, and Meteorology, Faculty of Mathematics, Physics and Informatics, Comenius University Bratislava, Mlynská dolina, 84248 Bratislava, Slovakia; kulish1@uniba.sk

5   Max-Planck-Institut für Radioastronomie, Auf dem Hügel 69, 53121 Bonn, Germany; pbenke@mpifr-bonn.mpg.de

6   Institut für Theoretische Physik und Astrophysik, Universität Würzburg, Emil-Fischer-Str. 31, 97074 Würzburg, Germany

7   Department of Astronomy, Institute of Physics and Astronomy, ELTE Eötvös Loránd University, Pázmány Péter sétány 1/A, H-1117 Budapest, Hungary

8   HUN-REN–ELTE Extragalactic Astrophysics Research Group, ELTE Eötvös Loránd University, Pázmány Péter sétány 1/A, H-1117 Budapest, Hungary

*   Correspondence: frey.sandor@csfk.org

**Abstract:** The prominent radio quasar PKS 2215+020 (J2217+0220) was once labelled as a new laboratory for core–jet physics at redshift $z = 3.572$ because of its exceptionally extended jet structure traceable with very long baseline interferometric (VLBI) observations up to a $\sim$600 pc projected distance from the compact core and a hint of an arcsec-scale radio and an X-ray jet. While the presence of an X-ray jet could not be confirmed later, this active galactic nucleus is still unique at high redshift with its long VLBI jet. Here, we analyse archival multi-epoch VLBI imaging data at five frequency bands from 1.7 to 15.4 GHz covering a period of more than 25 years from 1995 to 2020. We constrain apparent proper motions of jet components in PKS 2215+020 for the first time. Brightness distribution modeling at 8 GHz reveals a nearly 0.02 mas yr$^{-1}$ proper motion (moderately superluminal with apparently two times the speed of light), and provides $\delta = 11.5$ for the Doppler-boosting factor in the inner relativistic jet that is inclined within 2° to the line of sight and has a $\Gamma = 6$ bulk Lorentz factor. These values qualify PKS 2215+020 as a blazar, with rather typical jet properties in a small sample of only about 20 objects at $z > 3.5$ that have similar measurements to date. According to the 2-GHz VLBI data, the diffuse and extended outer emission feature at $\sim$60 mas from the core, probably a place where the jet interacts with and decelerated by the ambient galactic medium, is consistent with being stationary, albeit slow motion cannot be excluded based on the presently available data.

**Keywords:** quasars; radio continuum; high-redshift galaxies; active galaxies; jets

## 1. Introduction

Active galactic nuclei (AGN) are among the most powerful persistent emitters of electromagnetic radiation in the Universe, in the broadest range of wavebands, from radio to $\gamma$-rays (e.g., [1]). They are powered by accretion onto supermassive ($\sim$10$^6$–10$^{10}$ M$_\odot$) black holes (SMBHs) in the central region of galaxies. Radio-loud (i.e., with radio to optical flux density ratio $S_{5\,\mathrm{GHz}}/S_{4400\,\text{Å}} > 10$, [2]), sometimes also termed as "jetted" [3]

objects constitute the relative minority, nearly 10% of the general AGN population (e.g., [4]). They bear the most promising evidences of radio jets. However, observations have found occurrence of radio jets in radio-quiet AGN, too (e.g., [5–11]). The radio emission of radio-loud AGN is dominantly non-thermal and originates from charged particles accelerated to relativistic speeds in magnetic fields. Relativistic plasma jets are emanating from the close vicinity, launching from within ∼100 gravitational radii [12,13] of the central SMBH in opposite directions, perpendicular to the accretion disc. If one of these bipolar jets is viewed at a small inclination angle ($\iota$) to the line of sight, its radiation is enhanced by relativistic Doppler boosting, and we observe radio quasars or, in case of $\iota \lesssim 10°$, blazars [14,15].

Jetted or non-jetted, certain AGN are so luminous that they are observable from vast cosmological distances. The currently known record holder quasar is at the redshift of $z \approx 10$ [16], corresponding to only about 3.5% of the present age of the Universe. The redshifts of the most distant jetted quasars known to date are approaching $z \approx 7$ [17]. Despite recent discoveries, high-redshift ($z \gtrsim 4$) radio-loud quasars are still rare: there are about 300 of them known [18]. They are unique signposts for studying the onset of SMBH accretion in the early Universe, the formation of the first SMBHs, and they provide potential probes for testing cosmological world models.

Because of their compact, bright radio emission, jetted AGN are ideal targets for observations using the technique of very long baseline interferometry (VLBI) that provides the finest angular resolution imaging in astronomy (e.g., [19]). At GHz frequencies, VLBI is capable of imaging the jet structure at milliarcsec (mas) scale angular resolution, corresponding to ∼1–10 pc projected linear scales at high redshifts. Long-term VLBI monitoring observations of large samples of jetted AGN (e.g., [20–24]) allow us the study of the evolution of pc-scale radio structures. However, sources at high redshifts are less suitable for jet kinematic studies because *(i)* the cosmological time dilation makes changes happening in the rest frame of the AGN $(1 + z)$ times slower in the observer's frame, thus requiring longer monitoring time span for significant detection of motions, *(ii)* the observed frequencies correspond to $(1 + z)$ times higher emitted frequencies, where the optically thin steep-spectrum jet emission is weaker [25], and finally *(iii)* sources of similar luminosities naturally appear fainter when observed from large distances. All of these factors contribute to the fact that there are no dedicated VLBI jet kinematic surveys for high-redshift radio AGN, and the heterogeneous sample of jetted sources with jet proper motion measurements at $z \gtrsim 3.5$ contains only about 20 objects [26–30].

In the absence of dedicated long-term high-redshift AGN jet kinematic surveys, a useful approach is to analyse multi-epoch snapshot imaging data. These are available for the brightest sources that have been observed in astrometric/geodetic programs for decades [21,22]. Using such archival data supplemented by additional observations recently helped double the quasar jet proper motion sample at $z \gtrsim 3.5$ [30]. There are so few high-redshift jetted sources with jet kinematic measurements available at present that each new study is a potentially valuable addition to advance our understanding of the cosmological evolution of radio AGN. Are the physical properties of the high-redshift jets similar to those in low-redshift sources? Is there any cosmological evolution seen in jetted sources up to extremely large look-back times in the history of the Universe? These questions remain largely unanswered as of now. Moreover, the apparent jet component proper motion–redshift relation could be a valuable tool not only for jet physics but for cosmological studies as well (e.g., [31–33]), and in certain cases could even be applied for checking the validity of redshift measurements of AGN [34].

The subject of this paper, PKS 2215+020 (J2217+0220), is a powerful flat-spectrum radio quasar at $z = 3.572$ [35]. A more recent redshift determination from the Sloan Digital Sky Survey [36] is $z = 3.5909$, the optical spectrum with emission lines identified, can be seen at https://skyserver.sdss.org/dr17/en/get/SpecById.ashx?id=10322413208997 222400 (accessed on 2 February 2024). Quasi-simultaneous multi-colour optical and near-infrared photometric measurements of PKS 2215+020 were reported in [37]. The coordinates of this quasar in the International Celestial Reference Frame right ascension are RA =

$22^\text{h}17^\text{min}(48.2379348 \pm 0.0000051)^\text{s}$ and declination Dec $= 02°20'(10.71185 \pm 0.00011)''$ [38]. Its radio spectral index is $\alpha_{2.7\,\text{GHz}}^{5\,\text{GHz}} = -0.15$ [35] (following convention $S \propto \nu^\alpha$, where $S$ is the flux density and $\nu$ the observing frequency), and its rest-frame monochromatic power at 5 GHz is $P_{5\,\text{GHz}} \approx 2 \times 10^{28}\,\text{W}\,\text{Hz}^{-1}$. The quasar was a target of space-VLBI imaging observations at 1.6 GHz with the *HALCA* satellite of the Japanese VLBI Space Observatory Programme (VSOP) and a global array of 15 ground-based radio telescopes in 1997 [39]. The sensitive images revealed a uniquely large radio jet structure at such a high redshift, extending up to almost 80 mas or a ∼600 pc linear size projected onto the plane of the sky. (Throughout this paper, we assume a standard flat $\Lambda$ Cold Dark Matter cosmological model with Hubble constant $H_0 = 70\,\text{km}\,\text{s}^{-1}\,\text{Mpc}^{-1}$, matter density parameter $\Omega_\text{m} = 0.27$, and vacuum energy density parameter $\Omega_\Lambda = 0.73$. In this model, the luminosity distance of PKS 2215+020 is $D_\text{L} = 32351.8$ Mpc and the angular scale is $7.503\,\text{pc}\,\text{mas}^{-1}$ [40].) The long baselines from the ground-based radio telescopes to *HALCA* tripled the resolving power of the interferometer, making possible the measurement of the transverse size of the jet and the estimation of the mass of the central SMBH, approximately $4 \times 10^9\,\text{M}_\odot$ [39].

According to ground-only and space-VLBI images at 1.6 GHz, the morphology of the source is dominated by a bright compact (sub-mas) core that continues in a series of weak jet components, ending and also brightening up in the most extended (∼13 mas diameter) component, J1, at a ∼ 60 mas distance from the core [39]. Polarization-sensitive 5 and 8.4 GHz global VLBI imaging in 2001 [41] revealed a similar structure. The resolution of the ground-based 5 GHz VLBI image was comparable to that of the 1.6 GHz VSOP image [39], showing an equally extended core–jet structure, except for detecting the weak components in between the core and J1, insensitive to extended steep-spectrum jet emission. The bright region where the jet apparently changes direction from east to northeast (J1) was also seen in the 5 GHz global VLBI image, while it was resolved out at 8.4 GHz. Polarized emission was detected from the core only, with ∼1% degree of polarization [41].

The object is optically very faint ($m_\text{V} = 20.24 \pm 0.10$ [42]). PKS 2215+020 was first detected in X-rays with an unabsorbed (0.1–2.4) keV flux of $3.2 \times 10^{-13}\,\text{erg}\,\text{cm}^{-2}\,\text{s}^{-1}$ using the *ROSAT* High-Resolution Imager (HRI) instrument, and the data provided no clear evidence of arcsec-scale extended emission [43]. However, by comparing the same *ROSAT* HRI image with the 5 GHz radio image taken with the Very Large Array (VLA) interferometer at similar angular resolution, it was suggested that the ∼ 10' X-ray extension towards the northeast may be real and not caused by the asymmetric point-spread function of the HRI [39]. Notably, the arcsec-scale extension of the radio structure seen in the VLA image is in the continuation of the ∼10 mas-scale VLBI jet. Nevertheless, subsequent X-ray observations with the *Chandra* Advanced CCD Imaging Spectrometer (ACIS) found no sign of extension on the (3–10)' angular scale [44]. Assuming that the origin of the X-ray emission is the synchrotron self-Compton (SSC) process in the VLBI jet, there is mild relativistic beaming with Doppler factor $\delta \approx 1.5$ [44]. The SSC emission could be responsible for the extreme optical-to-X-ray spectral index $\alpha_\text{ox} = 0.384 \times \log(f_\text{o}/f_\text{x}) = 0.90$, where $f_\text{o}$ and $f_\text{x}$ are the rest-frame fluxes at 2500 Å and 2 keV, respectively [43]. The low value of the Doppler factor may indicate a large jet inclination with respect to the line of sight or a jet bending, preventing X-ray emission from being strongly beamed [44].

To reveal the physical and geometric properties of the radio jet in PKS 2215+020, here, we present an analysis of multi-epoch VLBI imaging data taken at multiple frequencies. We use archival data from observations spanning more than 25 years, from 1995 to 2020. While this quasar has been observed with VLBI earlier [39,41,45], this is the first study of its jet component proper motions, allowing us determination of the bulk Lorentz factor and the inclination angle of the inner sub-mas-scale section of the jet and revelation of the possibly stationary nature of the distant (∼60 mas) extended radio feature. The observations and data reduction are described in Section 2, the results are presented in Section 3 and discussed in Section 4, and our findings are summarised in Section 5.

## 2. Observations and Data Reduction

VLBI data at five different frequency bands (1.7, 2.3, 4.4, 7.4–8.7, and 15.3–15.4 GHz) taken at various epochs between 1995 and 2020 were analysed. Details of these observations are listed in Table 1. For the sake of simplicity, we collectively refer to the X-band (7.4–8.7 GHz) observations as 8 GHz, and the U-band (15.3–15.4 GHz) observations as 15 GHz hereafter.

With some exceptions, the calibrated visibility data were taken from the Astrogeo database (http://astrogeo.org/cgi-bin/imdb_get_source.csh?source=J2217%2B0220, see also http://astrogeo.smce.nasa.gov/cgi-bin/imdb_get_source.csh?source=J2217%2B0220, accessed on 29 December 2023) maintained by L. Petrov. The rest of the data were obtained in the framework of two recent observing projects, one with the European VLBI Network (EVN; project code RSF08, PI: S. Frey) [46] and another with the Very Long Baseline Array (VLBA; project code BM438, PI: K.P. Mooley) [47]. These latter two experiments targeted the radio-emitting binary AGN candidate PSO J334.2028+1.4075 and used our current object of interest, the quasar PKS 2215+020, as a nearby (within ∼1°) phase-reference calibrator source. Further details of these observations are given in respective references [46,47]. The visibility data from both projects were (re)calibrated in [46]. For our study, we used those data sets along with the others downloaded from the Astrogeo website.

Most of the data were taken by the VLBA of the U.S. National Radio Astronomy Observatory (NRAO). This array consists of eight antennas in the continental North America, Brewster (BR), Fort Davis (FD), Hancock (HN), Kitt Peak (KP), Los Alamos (LA), North Liberty (NL), Owens Valley (OV), and Pie Town (PT), one antenna in Hawaii, Mauna Kea (MK), and one in the U.S. Virgin Islands, St. Croix (SC). Other antennas from the EVN, Jodrell Bank Mk2 (United Kingdom), Westerbork (The Netherlands), Effelsberg (Germany), Medicina (MC, Italy), Onsala (O8, Sweden), Sheshan (Shanghai, SH, China), Nanshan (Urumqi, UR, China), Toruń (TR, Poland), Svetloe (SV, Russia), Zelenchukskaya (ZC, Russia), Badary (BD, Russia), and Hartebeesthoek (HH, South Africa) participated in one experiment. In Table 1, it is indicated with the corresponding two-letter station codes preceded by a minus sign if certain antennas from the VLBA were missing from the array in any given experiment.

Hybrid mapping with CLEAN deconvolution [48] and self-calibration [49] cycles were used to produce images of PKS 2215+020 in the DIFMAP software [50] in a standard way. To characterise the source brightness distribution quantitatively, we fitted the final self-calibrated visibility data with two-dimensional circular Gaussian model components [51].

**Table 1.** Details of VLBI observations analysed.

| Epoch (years) | $\nu$ (GHz) | Stations | $t$ (s) | IF $\times$ BW (MHz) | Project |
|---|---|---|---|---|---|
| 1995.535 * | 2.27<br>8.34 | VLBA | 178<br>184 | 4 × 4<br>4 × 4 | BB023 [45] |
| 1996.446 * | 15.36 | VLBA | 15,374 | 8 × 8 | BK042 [23] |
| 1998.426 * | 15.33 | VLBA | 17,888 | 1 × 64 | BG077 [23] |
| 2011.663 * | 8.36 | VLBA (−NL) | 135 | 8 × 16 | BC196 |
| 2013.561 * | 2.31<br>8.64 | VLBA (−BR, −FD, −MK) | 862<br>862 | 4 × 8<br>4 × 4 | RV100 |
| 2015.795 | 1.66 | JB, WB, EF, MC, O8, SH, UR,<br>TR, SV, ZC, BD, HH | 6379 | 1 × 128 | RSF08 [46] |
| 2016.000 * | 7.62 | VLBA | 242 | 8 × 32 | BP192 [52] |
| 2016.230 | 8.67 | VLBA | 18,383 | 2 × 107.5 | BM438 [46,47] |
| 2016.246 | 4.37<br>7.39 | VLBA (−MK) | 18,984<br>18,984 | 1 × 107.5<br>1 × 107.5 | BM438 [46,47] |

**Table 1.** *Cont.*

| Epoch (years) | $\nu$ (GHz) | Stations | $t$ (s) | IF $\times$ BW (MHz) | Project |
|---|---|---|---|---|---|
| 2016.364 | 15.37 | VLBA | 18,265 | 2 × 107.5 | BM438 [46,47] |
| 2016.548 * | 7.62 | VLBA | 244 | 8 × 32 | S7104 |
| 2016.657 * | 7.62 | VLBA (−PT, −SC) | 53 | 8 × 32 | BP192 [52] |
| 2018.349 * | 2.28 | VLBA | 110 | 4 × 32 | BS264 |
| | 8.65 | | 110 | 12 × 32 | |
| 2018.617 * | 2.28 | VLBA | 119 | 4 × 32 | BP222 |
| | 8.65 | | 119 | 12 × 32 | |
| 2020.882 * | 2.32 | VLBA † | 3515 | 4 × 16 | RV144 |
| | 8.64 | | 3515 | 4 × 16 | |

Notes: Col. 1: mean observing epoch; Col. 2: central observing frequency; Col. 3: participating VLBI antennas; Col. 4: on-source integration time; Col. 5: number of intermediate frequency channels times bandwidth; Col. 6: project code and reference (if available); * data obtained from the Astrogeo data base; † data from 3 other stations producing longer intercontinental baselines were discarded to achieve angular resolution comparable to other 8 GHz observing epochs.

## 3. Results

### 3.1. Core–Jet Structure at Multiple Frequencies

Our images show qualitatively the same core–jet structure as known from the literature for PKS 2215+020 [39,41]. Example images at all available frequencies are displayed in Figure 1. These data were taken in a relatively short time range (in years 2015–2020) compared to the full 25-year period covered by all archival observations (Table 1). Data sets with longer on-source integration and therefore the best sensitivities were selected. The image parameters are listed in Table 2. The size of the fields displayed is the same at each frequency to facilitate easy comparison.

Besides the compact core, an inner jet pointing towards the east can be seen in the zoomed-in higher-frequency images that provide sufficiently high angular resolution. In turn, an extended, diffuse emission region ∼60 mas east–northeast of the core is apparent in the wide field of the lower-frequency images that are more sensitive to the extended steep-spectrum emission. This region corresponds to the components designated with J1 and J2 in the earlier 1.6 GHz VSOP image [39] and can be traced here up to 8 GHz frequency (Figure 1).

Similarly to the 5 and 8.4 GHz VLBI images published earlier in the literature [41], our observations do not reveal the series of weaker components found at 1.6 GHz (J3 to J7, [39]). The reason is that our low-frequency VLBI images are less sensitive than the historical 1.6 GHz ground-based VLBI image [39], and at higher frequencies, the interferometer resolves out those weak, steep-spectrum jet features. However, because of the extended time coverage, we are able to follow and analyse the apparent proper motion of the inner jet component that could be detected in 10 epochs at 8 GHz, as well as the motion of a component resolved even closer to the core in three epochs at 15 GHz. Moreover, 2 GHz data in five epochs spanning 25 years allow us loose constraint of the apparent motion of the outer emission feature.

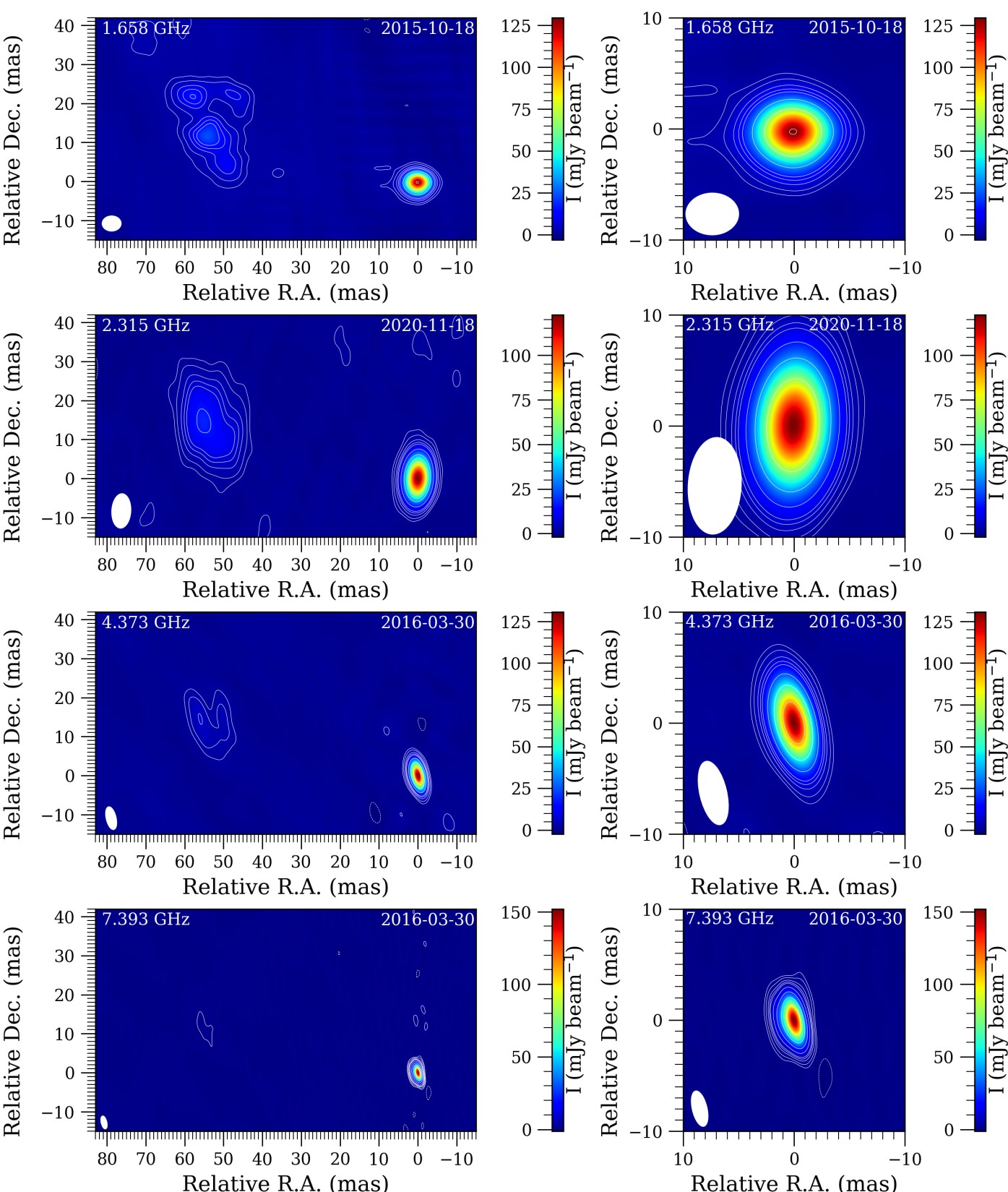

**Figure 1.** *Cont.*

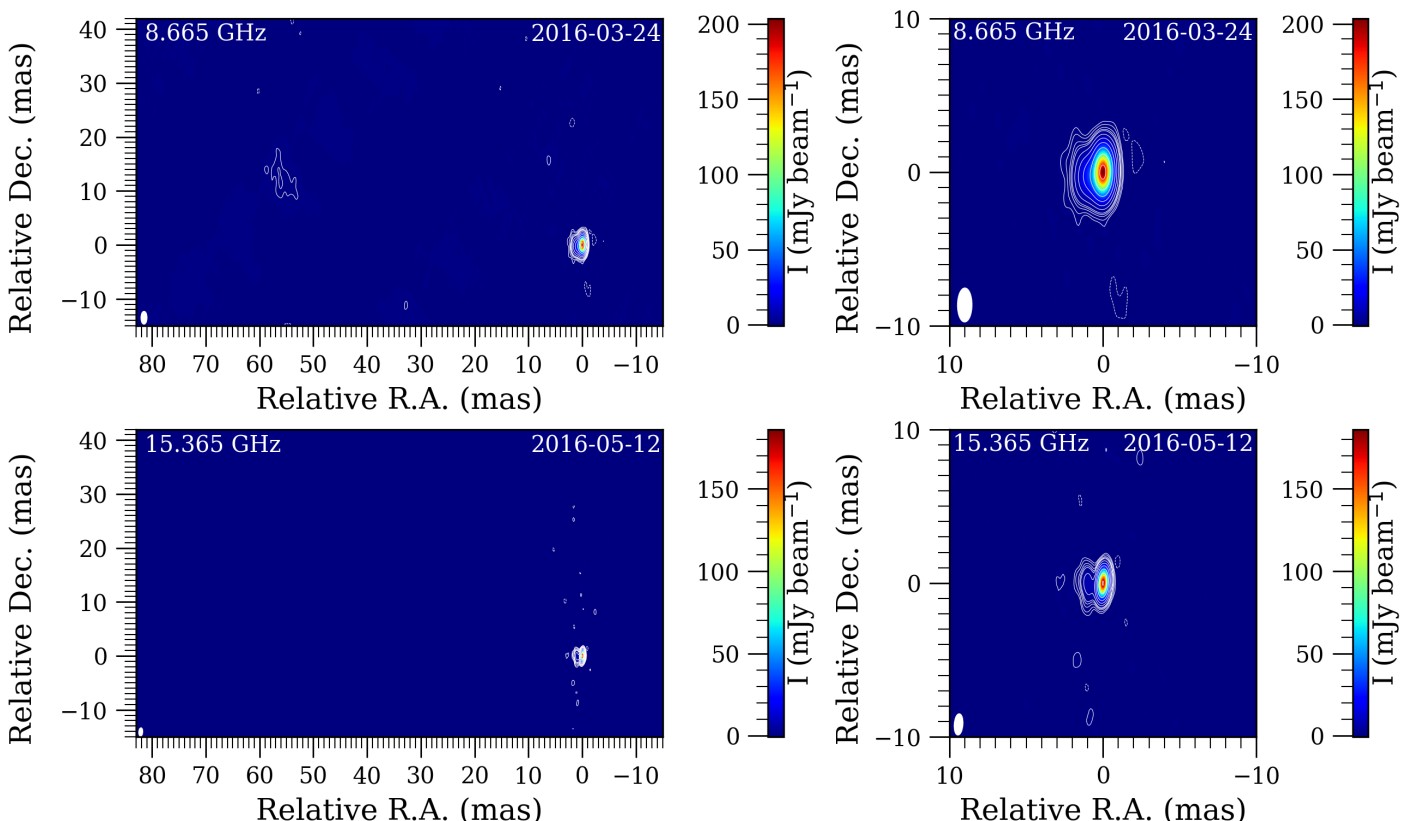

**Figure 1.** Selected naturally weighted VLBI images of PKS 2215+020 at six different frequencies, 1.7, 2.3, 4.4, 7.4, 8.7, and 15.4 GHz. Data sets with relatively long on-source integration and observing epochs close to each other in time (2015–2020) are chosen (see Table 1). The field of view in the left panels is set to be similar to that of the historical 1.6 GHz VSOP image [39] to facilitate easy comparison. In the right panel, the 20 mas × 20 mas field contains the core and the innermost jet section only. The images are centred on their brightness peak. The intensities are shown according to the colour scales on the right-hand side of the panels, as well as with the contours that start at around $\pm 3\sigma$ rms image noise, and the positive levels increase by a factor of 2. Negative contours are marked with dashed lines. The lowest contour level, peak intensity, and the elliptical Gaussian restoring beam parameters are given in Table 2. The restoring beams (FWHM) are also shown in the lower left corners. We note the gradual disappearance of the complex extended steep-spectrum feature towards the east at ∼60 mas that becomes resolved out as the frequency increases, and the central region being resolved into the flat-spectrum core and the innermost jet components at and above ∼7 GHz.

**Table 2.** Parameters of the VLBI images shown in Figure 1.

| $\nu$ (GHz) | $I_p$ (mJy beam$^{-1}$) | $I_0$ (mJy beam$^{-1}$) | $\Phi_{maj}$ (mas) | $\Phi_{min}$ (mas) | PA ($^\circ$) |
|---|---|---|---|---|---|
| 1.66 | 130 | $\pm 2.0$ | 4.69 | 3.69 | $-89.5$ |
| 2.32 | 123 | $\pm 1.0$ | 8.64 | 4.71 | $-3.3$ |
| 4.37 | 131 | $\pm 1.5$ | 5.77 | 2.36 | 13.2 |
| 7.39 | 152 | $\pm 0.7$ | 3.19 | 1.26 | 13.4 |
| 8.67 | 204 | $\pm 0.35$ | 2.12 | 0.86 | $-0.5$ |
| 15.37 | 186 | $\pm 0.35$ | 1.30 | 0.48 | $-4.1$ |

Notes: Col. 1—observing frequency; Col. 2—peak intensity; Col. 3—lowest intensity contour level (∼3σ image rms); Col. 4—restoring beam major axis (FWHM); Col. 5—restoring beam minor axis (FWHM); Col. 6—restoring beam major axis position angle, measured from north through east.

### 3.2. Inner Jet Component Proper Motion

Based on the circular Gaussian brightness distribution model components fitted to the self-calibrated visibility data at 8 GHz, we determined distance $r$ between the core (C) and the inner resolved component of the jet (JX). The model parameters obtained at 10 epochs are given in Table A1. From these measurements, we derived the apparent proper motion of the jet component as $\mu_{JX} = \mu = (0.0188 \pm 0.0005)$ mas yr$^{-1}$. The $r$ values as a function of time and the linear fit are displayed in Figure 2.

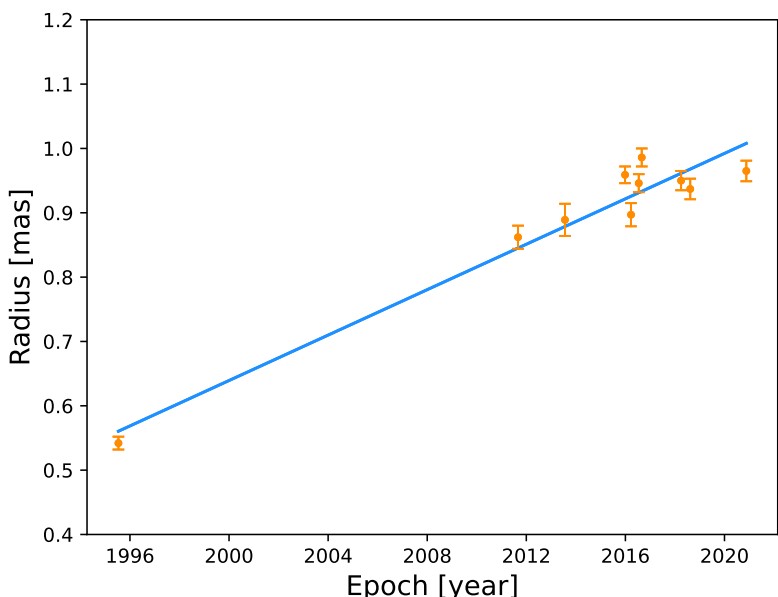

**Figure 2.** The core–jet distance as a function of time in PKS 2215+020 based on 10 epochs of VLBI observations and model fitting at 8 GHz. The slope of the fitted line produces apparent jet component proper motion $\mu = (0.0188 \pm 0.0005)$ mas yr$^{-1}$.

The time sampling of the 8 GHz VLBI measurements is far from being uniform. There is a large gap after the single earliest data point in 1995 (Figure 2). Therefore, this measurement has a decisive role in determining the magnitude of the proper motion. However, if we constrain our linear fit to the period after 2011, the proper motion obtained, $(0.011 \pm 0.003)$ mas yr$^{-1}$, is still significant, although with a larger error bar.

While the temporal coverage of the measurements is the best at 8 GHz (especially after 2010), there are three epochs between 1996 and 2016 with 15 GHz measurements also available, and the parameters of the innermost jet component (JU) can also be obtained (Table A2). We note that, because the angular resolution is better at this higher frequency, JU appears closer to the core than JX at 8 GHz. Nevertheless, based on 15 GHz data, the estimated linear proper motion $\mu_{JU} = (0.022 \pm 0.003)$ mas yr$^{-1}$ is consistent with the value of $\mu$ determined at 8 GHz within the uncertainties.

To express the apparent jet component proper motion ($\beta_{app}$) in the units of the speed of light ($c$), we can use the following formula (e.g., [53]):

$$\beta_{app} = 0.0158 \frac{\mu D_L}{1 + z}, \tag{1}$$

where $\mu$ is measured in mas yr$^{-1}$ and luminosity distance $D_L$ in Mpc. For jet component JX in PKS 2215+020, we obtain $\beta_{app} = 2.10 \pm 0.06$ apparent transverse speed, indicating superluminal motion.

### 3.3. Inner Jet Parameters

We use our 8 GHz observations to derive the inner jet parameters for PKS 2215+020. The brightness temperature values (Table A1) of the core component, i.e., the innermost unresolved section of the jet, are calculated as [54]

$$T_{\mathrm{b}} = 1.22 \times 10^{12} \, (1 + z) \frac{S}{\theta^2 \nu^2} \ \mathrm{K}, \tag{2}$$

where $z = 3.572$ is the redshift of the object, $S$ the flux density of the core in Jy, $\nu$ the observing frequency in GHz, and $\theta$ is the fitted Gaussian component diameter (full width at half maximum, FWHM) measured in mas. At all epochs, core sizes are larger than the minimum resolvable angular size of the interferometer [55].

The measured brightness temperatures, $10^{11} \, \mathrm{K} \lesssim T_{\mathrm{b}} \lesssim 2.5 \times 10^{12} \, \mathrm{K}$, all exceed the equipartition value, $T_{\mathrm{b,eq}} \approx 5 \times 10^{10} \, \mathrm{K}$ [56]; therefore, the radio emission of the jet is Doppler-boosted. In the following, we use the median of the 10 brightness temperature values, $T_{\mathrm{b,med}} = 4.7 \times 10^{11} \, \mathrm{K}$, to characterise the source. The median brightness temperature [57] is not sensitive to outliers caused by potential extreme source variability (we note that the median brightness temperature obtained from three measurements for the 15 GHz core is similar, $2.5 \times 10^{11} \, \mathrm{K}$).

The Doppler-boosting factor can be calculated from the ratio of the measured and intrinsic brightness temperatures,

$$\delta = \frac{T_{\mathrm{b,med}}}{T_{\mathrm{b,int}}} = 11.5, \tag{3}$$

where we adopt $T_{\mathrm{b,int}} = 4.1 \times 10^{10} \, \mathrm{K}$ found typical for a sample of jetted quasars that are not in outburst [57]. This value is very close to but somewhat lower than the commonly used $T_{\mathrm{b,eq}}$ [56].

Having derived the Doppler-boosting factor in the jet and the apparent speed of its inner resolved component, we can estimate the Lorentz factor ($\Gamma$) characteristic to the bulk motion of the plasma, as well as the jet inclination angle with respect to the line of sight (e.g., [14]):

$$\Gamma = \frac{\beta_{\mathrm{app}}^2 + \delta^2 + 1}{2\delta}, \tag{4}$$

$$\iota = \arctan\left(\frac{2\beta_{\mathrm{app}}}{\beta_{\mathrm{app}}^2 + \delta^2 - 1}\right). \tag{5}$$

We obtain $\Gamma = 5.99 \pm 0.01$ and $\iota = (1.77 \pm 0.02)°$ for PKS 2215+020.

### 3.4. Outer Jet Component Proper Motion

The outer jet feature is not resolved out and is clearly visible at 2 GHz (S band; see Figure 1) where VLBI observations are available in five different epochs from 1995 to 2020 (Table 1). We fit a single circular Gaussian model component (JS) to characterise this emission region (Table A3). This way, we can check whether its position is changing or not over a period of ~25 years. The fitted core–jet separation (~56 mas), the jet component position angle (~74°), as well as the component diameter (~11 mas) values are remarkably stable across all epochs. The formal radial proper motion of the JS component is $\mu_{\mathrm{JS}} = (0.03 \pm 0.08) \, \mathrm{mas \, yr^{-1}}$, i.e., consistent with this feature being stationary. However, its motion may formally be similar to that of the significantly detected values for JX and JU. We note that the flux densities of the fitted JS model components may in fact be more uncertain than suggested by formal errors (Table A3) if the complex and extended low surface brightness emission is not well represented by a simple model component.

## 4. Discussion

The small jet inclination angle determined for PKS 2215+020 for the first time clearly indicates that this AGN belongs to the class of blazars that are usually defined as having $\iota \lesssim 10°$ (e.g., [15]). The measured core brightness temperatures at 8 GHz (Table A1) are well over the equipartition limit [56]. The highest brightness temperature value, exceeding $2 \times 10^{12}$ K, is associated with a possible strong flux density outburst of the source in 1995. (Our method of using median $T_b$ for determining the Doppler factor is not sensitive to such outliers.) During outbursts, energy equipartition conditions between the particle and magnetic fields are not met [58]. Indeed, our fitted core component flux densities at both 2 and 8 GHz bands are remarkably, by factors of $\sim$3–7, higher in 1995 than in the later epochs (see Tables A1 and A3). According to instantaneous multi-frequency (1–22 GHz) total radio flux density monitoring [59] in recent years (from 2017 to 2020), this source was not strongly and rapidly variable during this period (https://www.sao.ru/blcat/, accessed on 29 December 2023). While PKS 2215+020 is classified as a flat-spectrum radio quasar, in fact, its power-law spectral index $\alpha$ derived from the broad-band radio spectrum [60] is slightly below $-0.5$, especially at lower ($\lesssim$1 GHz) observed frequencies, i.e., its radio spectrum was formally steep at the time of the study. However, radio power-law spectral index measurements can be affected by source variability and the frequency range where the index is calculated.

Earlier single-epoch 1.6 GHz VLBI imaging of PKS 2215+020 suggested that the outer ($\sim$60 mas) extended radio feature may be a working surface of the jet that interacts with the ambient medium of the host galaxy [39]. However, we are not aware of any observational information about the density of the interstellar medium (ISM) or its "clumpiness" in this galaxy which may influence the local interaction of the jet with the ISM. The scenario may be similar to that of the high-redshift ($z = 3.8$) radio AGN 4C 41.17, where a $\sim$10^8 $M_\odot$ structure was proposed to be responsible for a compact bright radio feature along the jet in a nearly kpc distance from the core [61]. In the case of another distant blazar, J0906+6930 at $z = 5.47$, an abrupt change in the projected jet direction, albeit on a smaller scale, was also explained by jet–ISM interaction [62]. There, the density of the ISM was estimated as $\rho_{ISM} \geq 2.4 \times 10^{-26}$ g cm$^{-3}$, corresponding to number density $n_e \gtrsim 27$ cm$^{-3}$, and the density contrast between the external medium and the jet material was found to exceed the value of nine. Simulations also show that the interaction of the jets with dense clouds can lead to the disruption of their propagation (e.g., [63] and references therein). With $n_e \sim 300$ cm$^{-3}$, jets can remain frustrated for millions of years. It is known that in compact steep-spectrum (CSS) radio sources [63], the jets are generally not powerful enough to pierce through the dense interstellar medium. However, PKS 2215+020 is not a CSS source, and we know from arcsecond-scale VLA imaging [39] that its jet extends to tens of kpc in projected linear size. Therefore, the density of the ISM here must be smaller.

Our multi-epoch 2 GHz VLBI data are not sufficient to detect proper motion of this component, supporting the notion that its radio emission is associated with a shock front, although its apparent proper motion may formally be similar to that of the inner jet components as well, just masked by the large error bar. Stationary components are often seen in VLBI monitoring of low-redshift jetted AGN (e.g., [24]), and not unprecedented at high redshifts, either [30]. However, maybe because of the much smaller sample of sources studied at $z \gtrsim 3.5$ to date, there are only a few examples. The projected linear distance of the component from the core is $\sim$420 pc. Taking into account the small inner jet inclination angle, even if the jet is bent by as much as $\sim$10° until it reaches that region, the deprojected distance is $\sim$2 kpc, which is comparable to the linear extent of medium-sized symmetric objects [64] that are misaligned (unbeamed) jetted AGN with symmetric hot spots on both sides of the core separated by $\lesssim$15 kpc. An alternative explanation for the stationary feature is that, because of its curved trajectory, the jet once again becomes more closely aligned with the line of sight at the location of the outer component than in the intermediate $\sim$10 mas scale regions closer to the core.

There are indications of jet bending on a smaller scale, within about 1 mas from the core, from our model fits at 8 and 15 GHz. On the one hand, the outward motion of the 8 GHz jet component (JX) is not strictly radial, its position angle changes northward by about 20° from 1995 to 2020 (Table A1). The position angle of the innermost jet component identified at 15 GHz (JU) also appears to change in a similar manner, although we have limited information, only three epochs of data available in a 20-year period (Table A2). On the other hand, we could fit two distinct model components to the jet at 15 GHz in 1996, and the position angle of the innermost one (JU) was significantly larger than that of the second, more distant component (JU2). All these measurements are consistent with a projected jet trajectory with a ∼120° position angle at ∼0.5 mas from the core, while the angle changes to ∼100° at ∼1 mas. Future high-resolution VLBI imaging with higher sensitivity could be able to reveal more of the wiggling pattern of the jet, hinted also by earlier sensitive 1.6 GHz observations [39].

X-ray observations with *Chandra* on 29 August 2002 [44] detected the quasar core, implying a Doppler factor of about 1.5. This value is small compared to our estimate, $\delta = 11.5$. The difference may partly be caused by variability, since the VLBI data we used for determining the median core brightness temperature, and thus the characteristic Doppler factor, were taken at epochs quite far away in time from the *Chandra* experiment. But most likely, the inclination angle of the bending jet is larger at the location where the X-rays are dominantly produced, causing less strongly beamed X-ray emission [44].

The moderately superluminal apparent jet speed ($\beta_{app} \approx 2$) and the rather small bulk Lorentz factor ($\Gamma = 6$) place PKS 2215+020 among the "average" jetted quasars at high ($z \gtrsim 3.5$) redshifts known to date. The Lorentz factor value can be considered low, since in lower-reshift sources, the distribution has a peak around 10 [57]. Using general relativistic magneto-hydrodynamic simulations and spectral energy distribution fitting to simultaneous multi-waveband data of about 40 bright low-redshift AGN, it was found that the Lorentz factor tends to be significantly higher for flat-spectrum radio quasars (FSRQs) ($20 \lesssim \Gamma \lesssim 50$) than for BL Lac objects ($6 \lesssim \Gamma \lesssim 20$) [65]. In this context, the jet of PKS 2215+020 is more similar to that of the modelled BL Lacs, i.e., relatively less powerful, weaker and slower. This may also play a role in the possible jet deflection by the ISM, as suggested by the possible presence of the stationary outer feature in the VLBI images at ∼60 mas from the core. For better constraining the apparent motion of that feature, a continuing long-term VLBI monitoring would be required at 2 GHz.

The small apparent proper motion ($\mu \approx 0.02$ mas yr$^{-1}$) that could be significantly detected with decades-long VLBI monitoring only is a result of a combined effect of the relatively slow jet with a moderate Lorentz factor, the small jet viewing angle, and the cosmological time dilation at this high redshift. The most up-to-date $\mu - z$ diagram for high-redshift jetted AGN is published in [30], where the point representing PKS 2215+020 is hardly distinguishable from the majority of other objects at $z \approx 3.5$.

The sample of jetted AGN with VLBI proper motion measurements at $z \gtrsim 3.5$, now including also PKS 2215+020, is limited to about 20 sources [26–30]. It is still not large enough to allow for meaningful statistical comparisons of jet properties with lower-redshift quasar samples. Even worse, the perspectives are not promising for a significant expansion of the high-redshift sample in the near future. First of all, the total number of jetted AGN known to date is relatively small in this redshift range. Moreover, among these few hundreds of sources, the majority has not been imaged with VLBI yet. But even if information is available on their mas-scale structure, most of these sources show compact and relatively weak (at best a ∼ 10 mJy level) radio emission without prominent extended jet structure (e.g., [66,67]). Therefore, these objects are unsuitable for jet kinematic studies. There are only a very few high-redshift sources with prominent jet structure, e.g., J0309+2717 at $z = 6.1$ [68], that were discovered recently. But follow-up VLBI imaging observations for at least two decades are needed to possibly detect significant component proper motions. Since these newly discovered objects are not bright enough to be included in regular astrometric/geodetic VLBI monitoring campaigns, like PKS 2215+020, in our case, the only

way to obtain jet kinematic information is to regularly revisit them with VLBI, preferably every couple of years, at a carefully selected observing frequency or frequencies.

## 5. Summary and Conclusions

Following up earlier single-epoch VLBI imaging studies [39,41], we revisited the high-redshift ($z = 3.572$) radio quasar PKS 2215+020 (J2217+0220) that is known for its unusually extended core–jet structure. We collected and analysed archival multi-epoch VLBI imaging data at five different frequency bands (1.7, 2.3, 4.4, 7.4–8.7, and 15.3–15.4 GHz) taken at several epochs between 1995 and 2020, but mostly after 2010. The images with $\sim$1–10 mas scale angular resolutions (Figure 1) show the bright compact core region resolved into distinct emission components at higher frequencies, and an extended, steep-spectrum feature $\sim$56 mas from the core at lower frequencies.

We were able to constrain the apparent proper motions of jet components in PKS 2215+020 for the first time. In particular, the jet component identified at 8 GHz in 10 epoch showed a moderately superluminal motion with apparent speed 2.1 times the speed of light. Its apparent proper motion was $\mu \approx 0.02$ mas yr$^{-1}$. Based on core brightness temperature measurements, we derived $\delta = 11.5$ for the characteristic Doppler-boosting factor, $\Gamma = 6$ for the bulk Lorentz factor, and $\iota = 1.8°$ for the jet inclination angle with respect to the line of sight. The latter value indicates that PKS 2215+020 belongs to the class of blazars. The factor of $\sim$5 variability in the 8-GHz core flux density between 1995 and the later epochs (Table A1) is also typical for blazars.

The outer emission feature could be modelled at 2 GHz in five epochs covering $\sim$25 years. We speculated that at this location the jet may interact with the ambient galactic medium which slows it down. Indeed, model-fitting results indicate no significant change in the angular distance of the component from the core ($\sim$56 mas) and its position angle ($\sim$74°), suggesting that this component may be stationary.

The small apparent proper motion value and jet parameters derived for PKS 2215+020 are typical for other known $z \gtrsim 3.5$ radio AGN with similar measurements available to date. However, this sample is currently very small, containing only about 20 sources, and not expected to be expanded significantly in the coming decade or two, either. This makes the addition of PKS 2215+020 potentially even more valuable. It is the right time to define a sample of high-redshift radio AGN with prominent jets known from the literature to initiate a dedicated long-term VLBI observing campaign designed for jet kinematic studies.

**Author Contributions:** Conceptualization, S.F.; methodology, S.F.; formal analysis, S.F., J.F., K.P., K.K., P.B., D.K.; software, S.F., J.F., K.P.; validation, S.F., J.F., K.P., K.É.G.; writing—original draft preparation, S.F., J.F.; writing—review and editing, S.F., J.F., K.P., K.K., P.B., D.K., K.É.G.; visualization, J.F., K.P.; supervision, S.F. All authors have read and agreed to the published version of the manuscript.

**Funding:** This research was funded by the Hungarian National Research, Development and Innovation Office (NKFIH), grant numbers OTKA K134213 and PD146947. This project received funding from the HUN-REN Hungarian Research Network. K.K. was funded by the International Astronomical Union—International Visegrád Fund Mobility Award, grant number 22210105. P.B. was supported through a PhD grant from the International Max Planck Research School (IMPRS) for Astronomy and Astrophysics at the Universities of Bonn and Cologne.

**Data Availability Statement:** The calibrated VLBI data are available from the Astrogeo archive (http://astrogeo.org/ and https://astrogeo.smce.nasa.gov/vlbi_images/ (accessed on 1 February 2024)) or from the corresponding author upon reasonable request.

**Acknowledgments:** We thank the three anonymous referees for their valuable comments and suggestions. The EVN is a joint facility of independent European, African, Asian, and North American radio astronomy institutes. Scientific results from data presented in this publication are derived from the following EVN project code: RSF08. The National Radio Astronomy Observatory is a facility of the National Science Foundation operated under cooperative agreement by Associated Universities, Inc. We acknowledge the use of archival calibrated VLBI data from the Astrogeo Center database

maintained by Leonid Petrov. D.K. is grateful for the support received from the observatory assistant programme of the Konkoly Observatory [69].

**Conflicts of Interest:** The authors declare no conflict of interest. The funders had no role in the design of the study; in the collection, analyses, or interpretation of data; in the writing of the manuscript; or in the decision to publish the results.

## Abbreviations

The following abbreviations are used in this manuscript:

| | |
|---|---|
| ACIS | Advanced CCD Imaging Spectrometer |
| AGN | Active galactic nuclei |
| CSS | Compact steep spectrum (radio source) |
| EVN | European VLBI Network |
| FSRQ | Flat-spectrum radio quasar |
| FWHM | Full width at half maximum |
| HALCA | Highly Advanced Laboratory for Communications and Astronomy |
| ISM | Interstellar medium |
| HRI | High-Resolution Imager |
| NRAO | National Radio Astronomy Observatory |
| ROSAT | Roentgen Satellite |
| SMBH | Supermassive black hole |
| SNR | Signal-to-noise ratio |
| SSC | Synchrotron self-Compton |
| VLBA | Very Long Baseline Array |
| VLBI | Very Long Baseline Interferometry |
| VSOP | VLBI Space Observatory Programme |

## Appendix A

In Tables A1–A3, we provide the parameters of the Gaussian brightness distribution model components fitted to the calibrated VLBI visibility data of PKS 2215+020 at 8, 15, and 2 GHz, respectively. We followed [70] in estimating the uncertainties of model parameters. For flux densities, we added an extra 5% error in quadrature to account for VLBI amplitude calibration uncertainties (e.g., [67,71]).

**Table A1.** Parameters of the circular Gaussian components fitted to 8 GHz data and the calculated core brightness temperatures.

| Epoch (years) | Comp. | $S$ (mJy) | $\theta$ (mas) | $r$ (mas) | $\phi$ (°) | $T_b$ ($10^{11}$ K) |
|---|---|---|---|---|---|---|
| 1995.535 | C | $702 \pm 43$ | $0.15 \pm 0.01$ | ... | ... | $25.0 \pm 3.3$ |
| | JX | $71 \pm 16$ | $0.52 \pm 0.02$ | $0.54 \pm 0.01$ | $124.4 \pm 1.0$ | |
| 2011.663 | C | $118 \pm 9$ | $0.23 \pm 0.01$ | ... | ... | $1.8 \pm 0.2$ |
| | JX | $48 \pm 3$ | $0.65 \pm 0.04$ | $0.86 \pm 0.02$ | $107.2 \pm 1.2$ | |
| 2013.561 | C | $104 \pm 9$ | $0.28 \pm 0.02$ | ... | ... | $1.0 \pm 0.1$ |
| | JX | $27 \pm 4$ | $0.69 \pm 0.05$ | $0.89 \pm 0.03$ | $102.6 \pm 1.6$ | |
| 2016.000 | C | $117 \pm 8$ | $0.19 \pm 0.01$ | ... | ... | $3.1 \pm 0.3$ |
| | JX | $26 \pm 2$ | $0.56 \pm 0.03$ | $0.96 \pm 0.01$ | $101.6 \pm 0.8$ | |
| 2016.230 | C | $200 \pm 11$ | $0.12 \pm 0.01$ | ... | ... | $10.3 \pm 1.7$ |
| | JX | $43 \pm 3$ | $0.73 \pm 0.04$ | $0.90 \pm 0.01$ | $106.1 \pm 0.9$ | |
| 2016.548 | C | $134 \pm 8$ | $0.13 \pm 0.01$ | ... | ... | $7.6 \pm 1.2$ |
| | JX | $29 \pm 2$ | $0.62 \pm 0.03$ | $0.95 \pm 0.01$ | $102.6 \pm 0.8$ | |
| 2016.657 | C | $143 \pm 9$ | $0.18 \pm 0.01$ | ... | ... | $4.2 \pm 0.5$ |
| | JX | $28 \pm 2$ | $0.66 \pm 0.03$ | $0.97 \pm 0.01$ | $102.9 \pm 0.8$ | |

**Table A1.** *Cont.*

| Epoch (years) | Comp. | $S$ (mJy) | $\theta$ (mas) | $r$ (mas) | $\phi$ (°) | $T_{\mathrm{b}}$ ($10^{11}$ K) |
|---|---|---|---|---|---|---|
| 2018.349 | C | $157 \pm 10$ | $0.19 \pm 0.01$ | ... | ... | $3.2 \pm 0.3$ |
|  | JX | $28 \pm 3$ | $0.54 \pm 0.03$ | $0.95 \pm 0.02$ | $101.6 \pm 0.9$ |  |
| 2018.617 | C | $153 \pm 10$ | $0.15 \pm 0.01$ | ... | ... | $5.1 \pm 0.7$ |
|  | JX | $23 \pm 2$ | $0.48 \pm 0.03$ | $0.94 \pm 0.02$ | $96.5 \pm 1.0$ |  |
| 2020.882 | C | $208 \pm 14$ | $0.15 \pm 0.01$ | ... | ... | $6.9 \pm 0.9$ |
|  | JX | $22 \pm 4$ | $0.69 \pm 0.03$ | $0.97 \pm 0.02$ | $101.5 \pm 1.0$ |  |

Notes: Col. 1—observing epoch; Col. 2—component identifier; Col. 3—flux density; Col. 4—diameter (FWHM); Col. 5—angular separation from the core (C); Col. 6—position angle of the component with respect to the core, measured from north through east; Col. 7—brightness temperature of the core.

**Table A2.** Parameters of the circular Gaussian components fitted to 15 GHz data.

| Epoch (years) | Comp. | $S$ (mJy) | $\theta$ (mas) | $r$ (mas) | $\phi$ (°) |
|---|---|---|---|---|---|
| 1996.446 | C | $250 \pm 14$ | $0.153 \pm 0.004$ | ... | ... |
|  | JU | $22 \pm 5$ | $0.251 \pm 0.008$ | $0.428 \pm 0.004$ | $126.8 \pm 0.5$ |
|  | JU2 | $12 \pm 1$ | $0.78 \pm 0.09$ | $0.93 \pm 0.05$ | $109 \pm 3$ |
| 1998.426 | C | $117 \pm 8$ | $0.26 \pm 0.01$ | ... | ... |
|  | JU | $34 \pm 5$ | $0.38 \pm 0.03$ | $0.44 \pm 0.01$ | $122 \pm 2$ |
| 2016.364 | C | $191 \pm 11$ | $0.112 \pm 0.004$ | ... | ... |
|  | JU | $17 \pm 1$ | $0.73 \pm 0.11$ | $0.96 \pm 0.06$ | $97 \pm 3$ |

Notes: Col. 1—observing epoch; Col. 2—component identifier; Col. 3—flux density; Col. 4—diameter (FWHM); Col. 5—angular separation from the core (C); Col. 6—position angle of the component with respect to the core, measured from north through east.

**Table A3.** Parameters of the circular Gaussian components fitted to 2 GHz data.

| Epoch (yr) | Comp. | $S$ (mJy) | $\theta$ (mas) | $r$ (mas) | $\phi$ (°) |
|---|---|---|---|---|---|
| 1995.535 | C | $537 \pm 39$ | $0.36 \pm 0.02$ | ... | ... |
|  | JS | $130 \pm 9$ | $12 \pm 3$ | $56 \pm 2$ | $74 \pm 2$ |
| 2013.561 | C | $150 \pm 10$ | $1.05 \pm 0.05$ | ... | ... |
|  | JS | $93 \pm 6$ | $11 \pm 1$ | $55.4 \pm 0.6$ | $74.4 \pm 0.6$ |
| 2018.349 | C | $163 \pm 12$ | $1.16 \pm 0.06$ | ... | ... |
|  | JS | $108 \pm 6$ | $12 \pm 2$ | $55.8 \pm 0.8$ | $74.4 \pm 0.8$ |
| 2018.617 | C | $142 \pm 11$ | $0.97 \pm 0.06$ | ... | ... |
|  | JS | $99 \pm 6$ | $12 \pm 2$ | $56.6 \pm 0.9$ | $74.2 \pm 0.9$ |
| 2020.882 | C | $128 \pm 10$ | $1.30 \pm 0.09$ | ... | ... |
|  | JS | $69 \pm 4$ | $11 \pm 2$ | $56 \pm 1$ | $75 \pm 1$ |

Notes: Col. 1—observing epoch; Col. 2—component identifier; Col. 3—flux density; Col. 4—diameter (FWHM); Col. 5—angular separation from the core (C); Col. 6—position angle of the component with respect to the core, measured from north through east.

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
