# Peer review of "Revisiting a Core–Jet Laboratory at High Redshift: Analysis of the Radio Jet in the Quasar PKS 2215+020 at z = 3.572"

_universe, doi:10.3390/universe10020097_

Round 1

Reviewer 1 Report

Comments and Suggestions for Authors

Introduction:

Page 1:

           Radio-loud, or rather jetted [2] objects 22 constitute the relative minority, nearly 10% of AGN [3].

Comment: These studies have reported jets in radio-quiet AGN. AGN constituting a minority (~10% AGN population), are the most

 https://ui.adsabs.harvard.edu/abs/2023ApJ...959..107S/abstract

https://ui.adsabs.harvard.edu/abs/2020A%26A...636A..64B/abstract

https://ui.adsabs.harvard.edu/abs/2021A%26A...655A..95S/abstract

https://ui.adsabs.harvard.edu/abs/2019A%26A...627A..53H/abstract

Define what is radio loud. It is better to phrase as,

Despite radio loud  (LGHz/L4400 A>10,) AGN constituting ~ 10% of the general AGN population, they bear the most promising evidences of radio jets. However, recent observations have found occurence of radio jets in radio-quiet AGN too [cite the aforementioned articles].

Page 2:

           Relativistic plasma jets are emanating from the vicinity of the central 25 SMBH in opposite directions, perpendicular to the accretion disc.

Comment:

You can say that Recent EHT observation of M87 has found that the jets originate <500 R_g (gravitational radius) See:

Janssen+21

https://www.nature.com/articles/s41550-021-01417-w

·      The subject of this paper, PKS 2215+020 (J2217+0220), is a powerful flat-spectrum 68 radio quasar at z = 3.572

Comment: What is the radio luminosity?

Mention the radio spectral index and the Eddington ratio if known.

Results:

Page 7 (Table 2)

·  Please add the uncertainties associated with FWHM values if possible.

Discussion:

Page 11:

·      Earlier single-epoch 1.6-GHz VLBI imaging of PKS 2215+020 suggested that the outer 238 ( 60 mas) extended radio feature may be a working surface of the jet that interacts with 239 the ambient medium of the host galaxy [26].

Comment: Could you comment on the density on the ISM?

In CSS sources, the radio jets are compact due to dense environment (https://ui.adsabs.harvard.edu/abs/2021A%26ARv..29....3O/abstract) and the jets in this source (although Flat spectrum) may not be powerful enough to drill through. However, this source being powerful enough, the density of the surrounding medium may play a role in the jet-bending?

·  General comment on discussion: The paper is lacking a comparison with theoretical prediction. For example, this paper https://ui.adsabs.harvard.edu/abs/2015MNRAS.453.4070P/abstract

provides a theoretical model of jet dynamics. In their model, the bulk Lorentz factor is 20-50 for FSRQ. Whereas for BL Lacs, it is about 6-20. I am curious on the authors opinion, where this object stands compared to the theoretical predictions.

Summary and Conclusions:

·      Page 12: “By fitting circular Gaussian brightness distribution model component to the visibility 309 data,”

      Comment: It is redundant and can be removed.

Reviewer 2 Report

Comments and Suggestions for Authors Referee Report on Revisiting a Core -Jet Laboratory...by Frey et al.     The optical properties of the featured object are important for classification and interpretation, but it's just said to be a quasar, and the reference given doesn't show the spectrum.  No mention is made of optical variability or polarization;  it's just erroneously stated that the radio properties make it a blazar.  Can we see a spectrum or get any more optical unformation?   The combination of derived jet parameters seems very unusual to me in and I see why perhaps.  They use a very unconventional (unless practice has changed totally since I stopped following this extremely closely) method of inferring the delta boost factor.   In the usual simplified parametrization of rectilinear motion the delta parameter can only approach the bulk relativistic motion gamma factor, yet here it is twice as high.   Eventually I got to the explanation.  They use a method which I've never seen before and which seems dubious to me.  At least the parameters derived from it can't be compared to those in the vast majority of the literature.  Perhaps the authors can direct me to sufficient  documentation vetting their method.    They simply take the ratio of brightness temp Tb to the Readhead value for the equipartition Compton limit in the jet frame.  Sometimes they seem to treat the resulting ratio like a value and sometimes like a limit.  They acknowledge that equipartition conditions are unlikely to obtain at peak brightness, then seem to assume it does, and use a novel median value for that in the observed frame, which though very nonstandard doesn't sound crazy.  >>>Standard references such as Begelman Blandford and Rees 1984 use a much higher Tb ~ 1e12 K as the max Tb in the jet frame for incoherent synch rad.  That value might not he attained in practice, but i see no reason to simply assume this source has Tb of the max under the equipartition  assumption.  I've never seen that done before, and if applied to famous superluminal agn, the resulting delta boost factor would be much higher than that derived from simple models for the kinematics.       A minor point is the authors seem to take the Tb limit from a restoration with a circular beam of some chosen diameter, whereas the true beam is highly elliptical.   I don't think that's correct practice.  I don't know how the adopted circular beam compares in size with the actual highly elliptical beam.   --     Next there is a conflation of "jet" with "relativistic jet," the latter being only a subset of the objects the community accepts as jets.  Originally and today the term "jet" was defined as a radio morphological property:  A long thin synchrotron ray suggestive of radial plasma motion is the gist of the def.   I think the most widely accepted definition for a radio jet comes from the 1984 Annual Reviews of Astron and Astrophysics by Bridle and Perley, Sec 1.2. Of the three requirements it's clear the first and third aren't met.     By contrast it IS satisfied far better by some "radio quiet" objects than by the present object, contrary to the text.  See for example the magnificent, non- relativistic  jet in the radio quiet Seyfert galaxy ngc4151:   https://ui.adsabs.harvard.edu/abs/2005AJ....130..936U/abstract   See fig 1.   So it's by no means correct to equate having a jet with a radio loud source, or a source with relativistic motion.  This radio quiet object has a jet if anything does, and this map isn't usual for Seyferts except in map quality.   --   >>>By contrast ironically the off-nuclear emissions in the present paper looks like anything but a jet, and doesn't come close to satisfying proposed definitions  or common usage, but seems somet8mrd to be referred to as part of the jet.   However given that the innermost off-nuclear component is said to move away from the identified core, i don't object to referring to it as a jet component if a slight qualification of the usage is requested.   The fact that the diffuse eastern, off - nuclear component shows flux changing by a factor of 2 suggests to me that the values are unreliable.   This is a lobe-like component hundreds of pc from the nucleus and unlikely to change that fast in my opinion. --   Sec 4   It's stated that the small estimated inclination of the innermost "jet" qualifies this as a blazar but again that isn't a definition I've seen anywhere.  Also the inclination estimate from assuming an equipartition jet rest frame maximum brightness temp is new to me and by no means accepted usage.     Highly beamed sources are also known for their rapid variability in the radio and it's stated that this object doesn't have that property. Similarly it's correctly noted that often a spectral index of greater than -0.5 is considered to indicate a "flat spectrum radio source" but that thus object doesn't qualify, though that's a rough rule and very sensitive to the freq used.    I will mention here that the sign convention for spectral index is used inconsistently.  When referring to alpha (o,x ) the negative convention is used.   I doubt the inferred tiny inclination of 1.8 degrees, and it's used incorrectly to de-project the source size.  They seem to assume a strictly 1-dimensional straight line for the morphology, inconsistent with the lobe size, and not reasonable.   Considering finite lobe size and the usual slight bending, extreme deprojection factors are unrealistic.   Regarding the maps, the core regions are pretty hard to see.  Also since radio maps like these substantial errors which do not manifest as just gaussian noise  in independent neighboring resolution elements, i think the readers need an essential clue on what to believe, that is, negative contours.  The images allow for -3 sigma contours, but they are almost non existent.  --   It's said that the x-ray observation yields a Doppler factor of (only) 1.5, but not how that is obtained or what the x-ray spectrum shows.  That doesn't support a blazar like interpretation but that's rationalized as down to variability or misorientation (?).   It's stated that the lack of detected motion of the 56 mas component indicates deceleration but no upper limit to its speed is given;  furthermore it's diffuse with low surface brightness so a sufficient tight upper limit would be hard to believe.     In sum, I think this paper needs to be cleaned up a lot, and when made more rigorous, it will be worth publishing all the nice data, though I'm not so familiar with this magazine to be sure it's the best place for it.      

Reviewer 3 Report

Comments and Suggestions for Authors

The paper by S. Frey et al. deals with he radio quasar PKS 2215+020 (J2217+0220) that has been identified as a potential laboratory for core–jet physics due to its extended jet on VLBI scale, up to 600 pc from the compact core. The object is also remarkable for its high redshift, z = 3.572. The authors analyzed archival spectra spanning 25 years, and were able to measure the apparent proper motion of jet components for the first time, with a moderately superluminal motion detected at 8 GHz.

The authors convincingly demonstrate the steep spectrum of the feature they identify at ~60 milliarcsec from the core. The interpretation of this feature as a region of stationary nature, with potential interaction with the ambient galactic medium appears to be well motivated.

Other interesting findings concern the Doppler-boosting factor, bulk Lorentz factor, and jet inclination angle, categorizing PKS 2215+020 as a blazar.

The paper reads well, is consequentially developed, and to-the-point. The unique property of the source as well as its high redshift should make the analysis of the authors of high interest to workers in the field. 

I have only one major concern about the results of the paper. Fig. 2 shows that their inference on the amplitude of the apparent proper motion is based on a correlation that is statistically biased (in practice supported by a single, outlying point, one of the configurations known in statistics  as "Anscombe quartet"). The remaining data points apparently do not show any correlation, likely because of the much shorter temporal  baseline. In addition, the authors do not show the data relative to the positional change from which they infer the superluminal change. 

So my suggestion is that the authors give at the very least a word of caution concerning the  fit of Fig. 2  and, more importantly, that they show the 1996 and a later map to compare and provide the evidence for their claim of superluminal motion. 

Comments on the Quality of English Language

English efine.

Round 2

Reviewer 3 Report

Comments and Suggestions for Authors

Thanks for addressing my concern.

Comments on the Quality of English Language

English is almost fine.